



# Development of Asia Pacific Weather Statistics (APWS) dataset for use in Soil and Water Assessment Tool (SWAT) simulations

Uttam Ghimire, Taimoor Akhtar, Narayan Kumar Shrestha, Prasad Daggupati

College of Engineering and Physical Sciences, University of Guelph, Ontario, N1G2W1, Canada

*Correspondence to* Prasad Daggupati ([pdaggupa@uoguelph.ca](mailto:pdaggupa@uoguelph.ca)), +1-519-760-9299

**Abstract.**

The application of Soil and Water Assessment Tool (SWAT) for hydrological modelling in Asia Pacific region is immense. However, a robust modelling practice is often constrained by limited amount and quality of weather data. In such conditions, SWAT uses an inherent statistical weather generator to generate synthetic series of weather inputs for which, long-term precise

weather statistics are needed. This study presents a high-resolution Asia Pacific Weather Statistics (APWS) dataset in a format ready to be used in SWAT simulations.

The APWS dataset consists of rainfall statistics from Asian Highly Resolved Observational Data Integration Towards Evaluation of Water Resources (APHRODITE) project at 0.25° and remaining weather statistics from Climate Forecast System Reanalysis (CFSR) at 0.38°. The utility of APWS is evaluated by comparing its performance with established CFSR statistics

for daily flow simulation in two river basins of South Asia; Narayani in Nepal and Wangchhu in Bhutan. The comparison is done on different precipitation data availability scenarios, where for each scenario, a specified percentage of historical precipitation data is removed and replaced by synthetic precipitation data, generated by SWAT's inherent weather generator with weather statistics from i) APWS and ii) CFSR independently.

The results indicated a clear outperformance of APWS over CFSR dataset in rainfall reconstruction, especially in the smaller

sub-basins. Statistics like probability of wet day following wet day, mean monthly rainfall and number of rainy days were found sensitive for better reconstruction of rainfalls series in the study river basins, inferring the advantage of using precise rainfall statistics. The APWS dataset is expected to contribute in better reconstruction of weather series needed for hydrological modeling using SWAT in the Asia Pacific region, and is publicly available at [https://hydra-water.shinyapps.io/APWS/](https://hydra-water.shinyapps.io/APWS/) or [http://doi.org/10.5281/zenodo.3460766](http://doi.org/10.5281/zenodo.3460766) (Ghimire et al., 2019).

Keywords: Asia Pacific, SWAT, APHRODITE, CFSR, Weather generator statistic

## 1 Introduction

The Asia Pacific region has been identified for its challenges in observed meteorological data quality and the sparse network of stations (Page et al., 2004;Martin et al., 2015;WMO, 2017), which has hindered robust agro-hydro modeling and climate risk assessments. In such data constrained regions, weather generators are potential options to generate synthetic series of



rainfall, temperature, humidity, and solar radiation (Semenov and Barrow, 1997). Weather generators are expected to reproduce the spatiotemporal dynamics of observed weather variables, their variability and persistence in a distribution (Ailliot et al., 2015). Their applications have been reported for energy demands (Kolokotroni et al., 2012), crop management (Supit et al., 2012), climate risk assessment (Steinschneider and Brown, 2013;Srivastav and Simonovic, 2015), agricultural (Jones and Thornton, 2013) and hydrological modelling (Dile and Srinivasan, 2014), among many others.

Importance of weather generators in hydrological modeling is paramount in data sparse basins (Candela et al., 2012;Dile and Srinivasan, 2014), either to generate a new series of weather inputs (Eames et al., 2012;Caraway et al., 2014) or to fill the missing and dubious information (Aouissi et al., 2016;Lu et al., 2015) in measured data. Of the several hydrological models employing weather generator for such purposes, Soil and Water Assessment Tool (SWAT) (Arnold et al., 2012) is, arguably, the most widely used. The application of SWAT for eco-hydrological modeling in Asia Pacific region has rapidly increased in

last few years and is further likely to increase with the on-going developments in SWAT (Francesconi et al., 2016;Arnold et al., 2012).

SWAT uses the WXGN weather generator (Sharpley, 1990) to generate or fill weather information using user specified statistics of rainfall, temperature, solar radiation, wind speed and dew point temperature (Aouissi et al., 2016). The WXGN weather generator is a statistical model that uses numerous core weather statistics (defined for each month) to generate

synthetic weather data. Of the total 168 monthly weather statistics needed to run WXGN in SWAT, 84 pertain to rainfall, highlighting the importance of rainfall statistics in weather generation (Neitsch et al., 2011). WXGN (also sometimes referred as WGN or WGEN) primarily generates the probability of rainfall occurrence for a given day and its corresponding amount, followed by other weather variables like temperature and solar radiation depending on the rainfall status (Richardson, 1981;Richardson and Wright, 1984). Thus, it is imperative that precise rainfall statistics must be defined for effective weather

generation and robust hydrological modeling in river basins, where rainfall is the primary component of hydrological cycle.

Currently, SWAT modelers have the option of manually providing weather statistics using observed weather data or using the publicly accessible Climate Forecast and System Reanalysis (CFSR) weather dataset (Saha et al., 2010), for hydrological simulation in basins located outside US (Neitsch et al., 2011). The SWAT development team has provided access to a few platforms to manually estimate the require weather statistics, e.g., "WGN Parameters Estimation Tool" and "WGN Excel

macro" (SWAT, 2019) etc. However, the amount of weather data required and the calculation procedures of the desired statistics, can be overwhelming for many SWAT modelers. Hence, most SWAT modelers prefer to use the already developed weather statistics CFSR dataset. CFSR dataset's extensive use for SWAT modeling has already been seen in developing countries around the world (Alemayehu et al., 2015;Dile and Srinivasan, 2014;Monteiro et al., 2016;Worqlul et al., 2017;Daggupati et al., 2017).

However, CFSR has been reported with higher biases in its weather variables compared to other gridded reanalysis products like MERRA, GLDAS, NCEP and ERA in various locations of North Western hemisphere (Decker et al., 2012). Even in Asian regions, CFSR has shown inferior performance in hydrological simulation in Three Georges Reservoir basin, China (Yang et al., 2014), Mekong region (Lauri et al., 2014), Srepok basin in Vietnam (Thom and Khoi, 2017), Maharlu lake in Iran (Eini et



al., 2019), Langcang basin in China (Tang et al., 2019) and many others, compared to other rainfall products. The resolution

of CFSR (0.38º) dataset could be another reason for its inferior performance in the topographically complex Asia Pacific region, as for each sub-basin, SWAT assigns the weather statistics from a nearby location defined within the dataset. As rainfall is the primary driver of hydrological models in majority of river basins of Asia, rainfall statistics defined from a location within 0.38º are likely to differ than that of sub-basin climatology and could yield deviations in reconstructing the rainfall and other weather series. Anders et al. (2006) reported that the rainfall differences within a 10 km spatial scale were as high as fivefold

in the Himalayan region. Such significant variations in rainfall characteristics are likely to impact the generation of better weather sequences and their applications for impact assessments.

Ideally, long term (more than 10 years) observed rainfall records at daily time-step are needed to define accurate rainfall statistics for the entire Asia Pacific region for better weather generation (Neitsch et al., 2011). However, the Asia Pacific region is sparsely gaged and long-term continuous weather records (WMO, 2017) are not publicly and readily available for many

gaged locations. The Asian Precipitation Highly Resolved Observational Data Integration Towards Evaluation (APHRODITE) gridded rainfall is a publicly available dataset that addresses the above-mentioned rainfall data availability challenge for the Asia Pacific region. APHRODITE is an interpolated product of thousands of surface rainfall stations from Asia Pacific countries and additional WMO Global Telecommunication Systems (Yatagai et al., 2012;Xie et al., 2007) that provides gridded rainfall data at a 0.25º spatial resolution (which is better than CFSR's 0.38º resolution). APHRODITE has been used in the

Asia Pacific region as baseline rainfall series for drought analysis (Um et al., 2017;Sohn et al., 2012), climate model assessments (Khan et al., 2018;Cruz and Sasaki, 2017), climate change impact assessments (Apurv et al., 2015;Kulkarni et al., 2013) and hydrological model setup (Lauri et al., 2014;Panday et al., 2014). Moreover, APHRODITE's relative superiority over other rainfall products, including CFSR, is well-established in several countries in the Asia Pacific region, including Saudi Arabia (El Kenawy and McCabe, 2016), Greater Mekong (Chen et al., 2017), Bhutan (Awange and Forootan, 2016), China

(Yang et al., 2014;Tang et al., 2019) and many others. The better performance of APHRODITE over CFSR and other products in the region suggests that rainfall statistics derived from APHRODITE data could be more precise, and hence, more effective in generating relatively accurate synthetic weather data and better flow simulations using SWAT in rainfall dominant basins of Asia.

Thus, the objecive of this study is two pronged; (1) development of a robust weather statistics dataset for effective weather

generation in river basins of Asia Pacific using APHRODITE rainfall to use in SWAT models and (2) evaluation of effectiveness of the proposed weather statistics dataset over existing CFSR dataset in weather generation and subsequent flow simulation in selected test basins. A high-resolution weather statistics dataset at 0.25º is generated (hereafter named APWS dataset, i.e., Asia Pacific Weather Statistics dataset) by combining rainfall statistics from APHRODITE and remaining weather statistics from nearest CFSR station at 0.38º spatial resolution and is made publicly accessible at https://hydra-

water.shinyapps.io/APWS/ or http://doi.org/10.5281/zenodo.3460766 (Ghimire et al., 2019) in SWAT ready format. Two river basins, Narayani in Nepal and Wangchhu in Bhutan are selected as test basins to validate the better performance of APWS over CFSR dataset in weather generation and flow simulation for different missing percentages of rainfall.



## 2 The APWS Dataset: Need, Preparation and Dissemination

### 2.1 The SWAT weather generator statistics data structure

SWAT is a semi-distributed hydrologic model that requires weather data input at the sub-basin level. Consequently, the weather generator embedded in SWAT (i.e., WXGN (Sharpley, 1990)) uses weather statistics inputs at the sub-basin level for generating synthetic weather data (if desired). Statistics for the weather generator are stored in SWAT's structured access database (i.e., SWAT2012.mdb for SWAT2012) Moreover, these statistics are stored for point locations (WXGN uses the

nearest location's statistics to generate synthetic weather data, wherever required), and should be derived from long term (more than 10 years) weather data (Neitsch et al., 2011).

While a default weather statistics data set (derived from US_First Order stations and US_COOP) is included in SWAT's default database for the United States (US), SWAT modelers who are interested in developing models for basins outside the US, need to manually provide weather statistics parameters. These parameters, along with their description and their effect on

weather generation using SWAT's WXGN weather generator are delineated in Table 1

*[Table 1 about here]*

### 2.2 Merits for the APWS dataset

The only data product that readily provides the weather statistics parameters (in SWAT-ready format) listed in Table 1, for point locations in the Asia Pacific region, is the CFSR weather dataset (SWAT, 2014). CFSR is a reanalysis data product (Saha

et al., 2010). Reanalysis data are generated (even in hind-cast scenarios) by performing data assimilation for a past period using historically available data from surface stations, satellites and airships and a current numerical weather prediction (NWP) model. For any pre-defined forecast (a hind-cast is used for generating weather statistics from CFSR) time period, NWP uses historical data (of the starting time of the hind-cast / forecast) as initial boundary condition of the atmosphere and generates the next first guess forecast (which for generating SWAT weather statistics, is a hind-cast) based on theoretical approximations

of atmosphere and relationship between different parameters (Parker, 2016). Consequently, accuracy of reanalysis-based hindcast datasets relies heavily on calibration of algorithms that represent the state of atmosphere. Given there is high uncertainty associated with the calibration of such algorithms, hindcast results of reanalysis-type data sets can have significantly higher uncertainty than weather dataset products that primarily rely on historical observations.

Numerous past studies show that reanalysis-type climate models have a tendency to over-estimate sea surface temperature

(Laprise et al., 2013), wind components (Brands et al., 2013), land temperature (Kim et al., 2014) and number of consecutive rainy days (more than 1 mm rainfall). Moreover, the effect of major cumulus parameterization closure scheme of climate models to simulate rainfall are found to largely affect the geographic distribution, frequency and intensity of rainfall (Qiao and





Liang, 2016). Qiao and Liang (2016) discussed that such closure schemes also tend to overestimate number of rainy days in rainfall scenarios of such models.

The overestimation tendency, especially for precipitation, is also prevalent in the CFSR dataset, for the Asia Pacific region (Hu et al., 2016). Hu et al. (2016) did a comprehensive analysis of multiple reanalysis precipitation datasets for Central Asia and reported that precipitation datasets based on spatially interpolated historical observations are more accurate than reanalysis-type data sets (including CFSR). Since, precipitation is a fundamental input for hydrologic models, it is imperative that if synthetic precipitation data (generated via weather generators) is used in hydrologic model development, this data is

produced via weather generators employing relatively accurate precipitation statistics (e.g., statistics 10-16 in Table 1). Hence, the focus of this study is on providing an alternate (to CFSR) weather statistics data set (in SWAT-ready format) for the Asia-Pacific region, where precipitation statistics are derived from observed historical data. This dataset, i.e., the Asia Pacific Weather Statistics (APWS) dataset, derives precipitation statistics from the APHRODITE data set (which is based on spatially interpolated historic data), and is described in detail in the next section.

**2.3 Preparation of APWS dataset**

The methodology adapted to generate the high-resolution dataset proposed in this study, i.e., the Asia Pacific Weather Statistics (APWS), is presented in Fig. 1.

*[Fig. 1 about here]*

APWS (see Fig. 1) derives rainfall statistics (at 0.25° resolution) from the historical observation-based APHRODITE dataset,

and other weather statistics from CFSR. Numerous past studies have shown that the APHRODITE dataset is effective for hydrologic modeling in the Asia-Pacific region (Lauri et al., 2014;Panday et al., 2014), and hence it is chosen for deriving rainfall statistics for APWS. For preparing the APWS dataset, APHRODITE rainfall data for the period 1981-2007 is accessed from http://search.diasjp.net/en/dataset/APHRO_PR, and extracted for each grid center (at 0.25° resolution) using customized scripts in R. The 27 year of rainfall data used in the study is expected to yield robust estimates of rainfall statistics, as suggested

by other studies (Fodor et al., 2013;Jones et al., 2010). The rainfall statistics, i.e., mean monthly rainfall (PCPMM), standard deviation (PCPSTD), skewness (PCPSKW), average number of rainfall days (PCPD), probability of wet day following dry day (PR_W(1,n)), probability of wet day following wet day (PR_W(2,n)) and half hour maximum rainfall (RAINHHMX), needed for the weather generator in SWAT (see Table 1) are then estimated at each of these grid centers on the APHRODITE rainfall data (Liersch, 2003), using the executable provided by SWAT creators, i.e., pcpSTAT.exe. PcpSTAT.exe is a Fortran

generated executable file provided by the SWAT development team to the potential SWAT modelers for the sole purpose of generating rainfall statistics using observed rainfall series (SWAT, 2019).

Since APHRODITE only includes rainfall statistics, remaining weather statistics of APWS that are needed in SWAT's weather generator (see Table 1), i.e., mean maximum temperature (TMPMX), mean minimum temperature (TMPMN), standard deviation of maximum temperature (TMPSTDMX), minimum temperature (TMPSTDMN), mean solar radiation

(SOLARAV), wind speed (WNDAV) and dew point temperature (DEWPT) are estimated from nearby CFSR locations, and



accessed from https://swat.tamu.edu/software/arcswat/. The Euclidean distance method is used to estimate the nearest CFSR stations for each grid center using customized R scripts. Finally, the hybrid weather statistics, which are collectively called APWS, are saved in an Excel file format which is compatible with SWAT's structured access database (that also includes weather statistics for SWAT's weather generator). The APWS dataset file has a size of approximately 50 MB and includes

statistics of 48,000 weather locations across Asia Pacific region (Fig. S1). Improvements of the proposed APWS dataset over existing CFSR dataset are better spatial coverage (0.25$^o$ in APWS vs 0.38$^o$ in CFSR) and precise rainfall statistics estimated from gridded observed rainfall data, compared to reanalysis data of CFSR. Section 3 provides a detailed illustration of how APWS has relatively superior performance over CFSR for hydrologic modeling in the Asia Pacific region under limited availability of precipitation data.

**2.4 APWS dissemination portal development**

Realizing the importance of ready access for finalized and SWAT usable weather statistics, a web application / portal is also created to easily access and filter the APWS statistics at country, basin or user defined levels. Figure 2 provides an overview of the interface of the APWS data access portal. As depicted in Fig. 2 ('Selection Panel' inside the interface), users of the portal may filter out statistics of a region of interest by either i) delineating a custom shape on the portal map (rectangle or

drawn polygon), ii) uploading a custom shape file, or iii) choosing a country. After selecting an area of interest, weather statistics of all data points within the area of interest may be downloaded as a csv file and subsequently imported into the SWAT database (i.e., the WGEN_user table) for use as weather generation statistics (Neitsch et al., 2011).

The APWS web-portal also has a basic visual analytics component that allows users to visualize time-series plots of rainfall and temperature statistics for selected grid centers of interest (that become active on the map, within the area of interest

selected; see Fig. 2). The APWS portal is developed in R, can be accessed from https://hydra-water.shinyapps.io/APWS/. The dataset can also be accessed from http://doi.org/10.5281/zenodo.3460766 (Ghimire et al., 2019).

*[Fig. 2 about here]*

**3 Performance evaluation of APWS dataset**

In order to evaluate the performance of APWS dataset in effective hydrological simulation using SWAT in the Asian region,

we used APWS for synthetic weather data generation for SWAT models of two river basins; Narayani (NRB) in Nepal and Wangchhu (WRB) in Bhutan (discussed in Sect. 3.1). Figure 3 provides an overview of the design of our performance evaluation experiment. We first develop, calibrate and validate SWAT models of the Narayani and Wangchhu basins (see Sect. 3.1-3.3). The calibrated SWAT models use historical rainfall records at multiple stations during model development and calibration. In order to assess the performance of precipitation statistics of APWS (against the default CFSR dataset used in

SWAT) in weather generation, we develop alterations (also called 'missing precipitation data' scenarios) of the historical precipitation dataset where, in each scenario a specified percentage of historical data is missing (the missing days are randomly



selected; discussed in Sect. 3.4). The SWAT models are then run with 'missing precipitation data' scenarios using i) APWS and ii) CFSR statistics (to generate synthetic precipitation records for missing precipitation days; also called reconstructed rainfall) and hydrological simulations using the reconstructed rainfall records are compared. The flows simulated using

reconstructed rainfall (from APWS and CFSR) are compared (see Sect. 3.5 for details) with flows simulated with unaltered rainfall (i.e., 0% missing data), as presented in methodological framework of Fig. 3. Finally, the sensitivity of rainfall statistics for precise hydrological simulation is assessed in terms of NSE and PBIAS (see Sect. 3.6).

*[Fig. 3 about here]*

### 3.1 Data acquisition for selected basins

The required rainfall, temperature and flow observations at daily timestep are acquired through Regional Integrated Multi Hazard Early Warning Systems (RIMES) center, Thailand and National Center of Meteorology and Hydrology (NCHM), Bhutan. Two river basins, Narayani (hereafter named NRB) in Nepal (36,000 sq.km) and Wangchhu (hereafter named WRB) in Bhutan (3,600 sq. km) are considered in this study to compare performance of APWS and CFSR statistics in weather generation. The location of NRB and WRB in south Asia, along with their topographical information and the flow stations

considered in this study is presented in Fig. 4.

*[Fig. 4 about here]*

The NRB consists of 79 rainfall, 36 temperature and 3 flow stations as presented in Fig. S2. Similarly, the WRB has 7 rainfall, 7 temperature and 3 flow stations, as presented in Fig. S3. The meteorological and flow data are available for the years 2008-2014 in the NRB and 2000-2014 in the WRB respectively.

### 210 3.2 Comparison of rainfall normals

While the focus of this study is on analyzing the effectiveness of APWS in hydrologic modeling for the Asia Pacific region using SWAT, we have also compared rainfall normals of the APHRODITE (used to develop precipitation statistics of our APWS dataset) and CFSR datasets for the NRB and WRB catchments. The purpose here is to directly compare the quality of the two datasets against historical precipitation observations.

A comparison of monthly cumulative rainfall amounts, their distribution and seasonality is done for selected rainfall stations in the NRB and WRB study basins. A common time period is established to compare the distribution and seasonality of precipitation at rainfall stations with APHRODITE and CFSR datasets. For WRB, 1981-2007 is chosen to compare the gridded (i.e., APHRODITE) and reanalysis (i.e., CFSR) rainfall series with observe rainfall. For NRB, observed rainfall data at all stations is only available for 2008-2014, thus the comparison is done for rainfall stations of Koshi river basin (i.e., another

river basin in Nepal with climate attributes similar to NRB) with that of corresponding APHRODITE and CFSR datasets for the 1981-2007 period.





### 3.3 Hydrological model setup

The SWAT model setup for NRB and WRB includes a 90mx90m digital elevation model (DEM) required for terrain processing and basin delineation (accessed from HydroSHEDS website https://hydrosheds.cr.usgs.gov/dataavail.php), a 300mx300m land
cover information (accessed from European Space Agency (ESA) Globcover project website http://due.esrin.esa.int/page_globcover.php), and a 1:5.000.000 scaled digital soil map of the world (DSMW) used to characterize soils in the study basins (accessed from https://worldmap.harvard.edu/data/geonode:DSMW_RdY). Since SWAT is a semi-distributed hydrologic model, the modeled basins are divided into sub-basins, and further into unique land units, also called Hydrologic Response Units (HRUs) based upon a combination of slope, land use and soil information. Since SWAT is
a highly parameterized model, both the NRB and WRB SWAT models are also calibrated a multi-site calibration. The SWATCUP software is used for calibration of both models, and the embedded Sequential Uncertainty Fitting (SUFI2) algorithm (Abbaspour, 2013) is used to optimize SWAT parameters to yield the best Nash Sutcliffe Efficiency (NSE).

### 3.4 Missing precipitation data scenario generation

Since a primary premise of this study is to compare the performance of the proposed APWS weather generation statistics
dataset against the CFSR statistics dataset for SWAT models developed for the Asia Pacific region, our dataset quality comparison experiment setup is based on generation of hydrologic modeling scenarios where precipitation records are missing (as depicted in Fig. 3). Eleven different precipitation scenarios are generated in this experiment setup where different percentages of rainfall data (i.e., 1, 5, 10, 15, 20, 25, 30, 35, 40, 45 and 50 percent) are missing from the historical rainfall record time-series. For each scenario, say $X$-% missing data scenario, precipitation records of X% days are randomly (uniform)
sampled (for each year of data record) and removed from the historical data set (Note: For each missing day, data of all rain gauges was removed from record). Since each precipitation scenario (say $X$-% missing data scenario) is stochastic, $N$ different instances (N = 100 in our experiments) are generated for each scenario.

Consequently, SWAT models with i) APWS and ii) CFSR weather statistics are run for all scenarios and instances. The WXGEN weather generator built in SWAT is automatically invoked to fill missing rainfall values. Hence, when the APWS
and CFSR weather statistics are applied in separate SWAT runs, for the same missing data scenarios and instances, we obtain separate hydrologic outputs (based on precipitation data filled by the weather generator using the different statistic sets). The hydrologic outputs generated via APWS and CFSR are subsequently compared against the 'baseline' SWAT hydrologic output, i.e., without any missing historical precipitation records. The criteria for quantifying the difference between APWS-based & CFSR-based hydrologic flows (under different missing precipitation scenarios), and baseline flows, are discussed in
Sect. 3.5.





## 3.5 Performance comparison

To provide equal weightage for low and high flows, the hydrologic flow values in this study are transformed initially using a Box-Cox transformation technique (Box and Cox, 1964) and then evaluated using the standard indices like Nash Sutcliffe Efficiency (NSE). Percentage Bias (PBIAS) is also used as a metric for comparing performance of the two weather statistics.

For the transformation, a lambda value of 0.25 is assumed following Willems (2009). The NSE and PBIAS metrics are computed by comparing i) flows simulated from reconstructed rainfall using APWS and CFSR datasets (under different missing precipitation scenarios discussed in Sect. 3.4) with ii) unaltered rainfall simulated flows (also called baseline flows as discussed in Sect. 3.4.

## 3.6 Sensitivity assessment of rainfall statistics

Sensitivity analysis aims to measure the impact of fluctuations in parameters of a model to its outputs or performance (Balaman, 2018). This study also aims to assess the sensitivity of synthetically generated precipitation data (via the SWAT weather generator) to the rainfall statistics used in SWAT's weather generator, i.e., PCPMM, PCPSTD, PCPSKW, PR_W(1,n), PR_W(2,n), PCPD and RAINHHMX. The sensitivity assessment mechanism is initiated by first creating 100 random missing precipitation scenarios (see Sect. 3.4 for description) of 30% missing rainfall data. The SWAT weather generator, i.e.,

WXGEN, is then used to generate precipitation data for the 30% missing days (for all 100 random scenarios), with the original APWS statistics dataset, and for selected precipitation stations in WRB, and subsequently the SWAT hydrologic model is run to generate simulated flows. The difference in simulated flows from generated weather data and actual weather data, i.e., without missing precipitation days (computed for each random scenario and quantified via NSE and PBIAS metrics) is recorded as the baseline / unaltered hydrologic performance of weather statistics. Subsequently, individual rainfall statistics

(i.e., PCPMM, PCPSTD, PCPSKW, PR_W(1,n), PR_W(2,n), PCPD and RAINHHMX) are changed by ±5, ±10 and ±25%, with one-at-a-time (OAT) approach (Cacuci et al., 2005), keeping other statistics fixed at their nominal values, and WXGEN is used to generate precipitation data for the 30% missing days (for all 100 random scenarios) with these altered rainfall statistics and used to drive the SWAT model for daily flow simulation. The sensitivity of rainfall statistics is finally shown in terms of Box-Cox transformed NSE and PBIAS indicators. These indicators are estimated for each OAT-altered rainfall

statistic scenario, by comparing i) simulated flows from generated weather data (from OAT-altered rainfall statistics) and ii) simulated flows obtained after running SWAT with actual weather data.

## 4 Results of APWS evaluation

## 4.1 Baseline rainfall comparison

A preliminary comparison of observed, APHRODITE and CFSR rainfall series for selected stations in the study basins

suggested that CFSR significantly differs from the observed rainfall (see Fig. 5). Although differences in median rainfall are





not significant at all stations, the distributions of monthly rainfall, depicted by violin-plots (over monthly precipitation data for years 1981-2007) in the top two rows of Fig. 5, illustrate that APHRODITE is more accurate, and thus more suitable, than CFSR for hydro-meteorological applications in the study basins. Similar outperformance of APHRODITE over CFSR is reported for other areas in the Asia Pacific region, e.g., Mekong (Lauri et al., 2014;Thom and Khoi, 2017), middle east (Eini

et al., 2019;Sidike et al., 2016) and China (Tang et al., 2019) and many others.

*[Fig. 5 about here]*

The bottom two rows of Fig. 5 show plots of mean monthly rainfall (observed, APHRODITE and CFSR) for selected stations of the study basin. These plots illustrate that APHRODITE data is consistent with observed monthly rainfall distribution, albeit exhibiting some underestimations. The relatively better performance of APHRODITE over CFSR data series is not surprising,

as the former is generated from the interpolation of ground rain gauges (Yatagai et al., 2012), while the latter is mostly a combination of satellite and observed data (Saha et al., 2010). The dominant rainfall seasonality (bottom two rows of Fig. 5) is also simulated well by APHRODITE, while CFSR has significant discrepancies in the study basins, which is expected to impact the hydrological application of CFSR statistics. A baseline comparison of mean annual rainfall computed using weather statistics of APHRODITE and CFSR shows significant differences in the volume of rainfall, mostly in the South East Asian

and Pacific countries, as presented in Fig. 6.

*[Fig. 6 about here].*

### 4.2 Hydrological model assessment

As mentioned in Sect. 3.3, SWAT models for the study basins, i.e., NRB and WRB, are developed and subsequently calibrated using SWAT-CUP. The calibrated SWAT models are able to simulate daily flows at most of the observed locations in study

basins with good accuracy (see Fig. 7). Nash Sutcliffe Efficiency (NSE) and Percentage Bias (PBIAS) are the metrics chosen to assess the daily flow simulation accuracy of SWAT for WRB and NRB. The NSE and PBIAS values computed from simulated and observed daily flows for each year in selected flow stations, representing top (Jomsom (NRB), Haa (WRB)), middle (Sisaghat, Damchhu) and lower parts (Devghat, Chimakoti) of the study basins are presented in Fig. 7. Results, as shown in the heat-maps of Fig. 7, reveal that that the calibrated SWAT models are able to capture variations in daily flows

with reasonable accuracy (as depicted by NSE metric values). Moreover, volumetric error between simulated and observed flows are also reasonable during both calibration and validation periods (depicted by PBIAS values).

*[Fig. 7 about here]*

The consistency of model in simulating flows with satisfactory accuracy was observed for individual years, as can be seen in Fig. 7. Generally, the performance of SWAT in upper parts of basins is relatively less accurate than in the middle and lower

parts. This trend has also been reported in other studies (Poncelet et al., 2017;Van Esse et al., 2013).



### 4.3 Performance of weather statistics: APWS and CFSR

Sections 3.4 and 3.5 describe the experimental setup employed in comparing performance of the APWS and CFSR datasets, for hydrologic modeling using SWAT in scenarios where observed precipitation is unavailable/missing. Eleven such scenarios

are compared, where *X-%* precipitation data is missing from historical records, and subsequently, the missing data is filled using statistics from APWS and CFSR statistics. Finally, SWAT models are run (with precipitation data filled using SWAT's weather generator using APWS or CFSR statistics), and differences in SWAT model outputs of the missing-precipitation scenarios and baseline scenario (with no missing data) are compared for APWS and CFSR.

The difference in hydrologic output (presented as NSE and PBIAS; the lines represent average NSE and PBIAS values over

multiple missing precipitation data scenarios (for the same percentage missing data) and the shaded areas represent standard deviation) using either APWS or CFSR, at flow locations of WRB is presented in Fig. 8. Results for WRB stations clearly show that accuracy of weather statistics is of paramount importance in filling the rainfall series and subsequently, in accurate simulation of hydrologic flows. The APWS dataset clearly outperforms CFSR in this regard, since both NSE and PBIAS values for APWS remain reasonable even with 50% missing precipitation data. Moreover, the difference in performance of the APWS

and CFSR statistics is more significant in smaller sub-basins located in upper parts of the basin (e.g., represented by the Haa station in left-most panels of Fig. 8), compared to the lower parts. A reason for this difference could be the subsequent dampening of the missing rainfall events, as the flow progresses downstream. The smaller sub-basins located in the upper parts of the study basins are more flashy in nature compared to the lower sub-basins, which has been established to negatively impact the hydrological model performance (Poncelet et al., 2017). The performance of hydrological models is also generally better

at the downstream locations and increases with size of basins (Merz et al., 2011;Van Esse et al., 2013).

*[Fig. 8 about here]*

A significant deviation of NSE is observed for all flow stations in WRB, when CFSR weather statistics are used to fill the missing rainfall series in the WRB. The rainfall statistics of CFSR are significantly different than observed and APHRODITE data for the basin, as evident from Fig. 5. This is likely to yield large errors from the baseline simulated flows (i.e., without

missing precipitation data) when CFSR is used to fill the missing rainfall series. The biased nature of CFSR in WRB is also evident from the PBIAS computed at its flow stations. The biases aggregate more than 50% in all stations, when 20% or less rainfall data is missing, and the weather generator with CFSR statistics is used to generate synthetic rainfall data for missing days.

*[Fig. 9 about here]*

Figure 9 compares the effectiveness of APWS and CFSR data sets in filling missing precipitation data for hydrologic modeling (using SWAT) of NRB. The relative superiority of the APWS dataset is also evident here. Results of NRB, as presented in Fig. 9, also depict that the size and location of a sub-basin has a significant impact on performance of weather statistics in simulating hydrologic flows, i.e., smaller sub-basins that are located in upper parts of basins, and have no contributions from other tributaries, tend to be heavily reliant on accurate observed weather data for accurate hydrologic simulation.





The Jomsom hydrological station located in the northernmost part of the NRB (Fig. 4, left-most panels) is part of a small sub-basin of NRB and is devoid of contribution from other tributaries in the basin. Moreover, the sub-basin that Jomsom drains has an arid climatology, with a mean annual rainfall of around 350 mm (as presented in Fig. S4). Arid basins have been known to have lower model efficiency compared to wet basins (Poncelet et al., 2017). Hence, the NSE and PBIAS values at Jomsom, become significantly worse (compared to the other stations located in lower parts of the basin), as the percentage missing data

value increases slightly. In arid and semi-arid sub-basins, total rainfall is mostly contributed by rainfall events rather than rainfall seasonality, due to which even a smaller percentage of missing data concentrated around such events is likely to deteriorate the hydrological model performance. The use of weather generator to reconstruct the missing rainfall is thus likely to change the rainfall sequence in such basins thus degrading the performance of weather generators significantly even for few missing events. Similarly, as the size of basin increases and as we approach the lower parts of NRB where rainfall volume is

significant, the reduction in performance of the weather statistics is gradual. Both APWS and CFSR datasets perform adequately in NRB for stations located in the lower part of the river basin (see results for Sisaghat and Devghat in Fig. 9). However, performance of APWS is slightly better for these stations as well. Overall, a consensus could be derived from both study basins that performance of APWS over CFSR statistics is better, in terms of deriving synthetic rainfall data for missing days at observed weather stations and, subsequently, in simulating hydrologic flows under limited precipitation data

availability scenarios.

## 4.4 Sensitivity of the rainfall statistics

The sensitivity analysis of rainfall statistics of APWS done over WRB in Bhutan, using One-at-a-time technique (see Sect. 3.6 for description of sensitivity analysis experiment), suggests that probability of a wet day following wet day (PR_W(2, n) is most sensitive in altering the performance of daily flows simulated using reconstructed rainfall (see Fig. 10 for results). Mean

rainfall followed by number of rainy days (PCPD) are found to be second and third most sensitive rainfall statistics for precise rainfall generation in WRB, as presented in Fig. 10. This is expected as the identification of a day as rainy/non-rainy depends upon the probability values defined in the weather statistics. Only when the day is designated rainy, the weather generator makes use of the mean, standard deviation and skewness of monthly rainfall to estimate the rain amount. Similarly, probability of wet day following dry day (PR_W (1, n)) is expected to have less sensitivity in the WRB basin, as generation of rainfall

values on a spell of dry days is unlikely to change the flow regime. Similarly, RAINHHMX is found insensitive to weather generation in the basin. The sensitivity of rainfall statistics on daily flow simulation is expected to vary with different climatology and basin characteristics.

*[Fig. 10 about here]*

A similar sensitivity analysis of the rainfall statistics in specifying model performance in terms of NSE also confirms that

PR_W(2,n), PCPMM and PCPD are most sensitive in rainfall reconstruction for the study basin (see Fig. S5).



## 5 Data availability

The AWPS dataset is archived for long-term storage and visual analytics at https://hydra-water.shinyapps.io/APWS/. The file size of Excel dataset is around 50 MB, which is itself in SWAT-ready format and can be accessed from http://doi.org/10.5281/zenodo.3460766 (Ghimire et al., 2019).

## 380 6 Conclusions

SWAT is a semi-distributed hydrologic model that is immensely popular in the Asia-Pacific region. However, given its semi-distributed nature, accuracy of SWAT is reliant on precipitation input data (that should be available at relatively high spatial and temporal resolutions). Since, availability of observed precipitation data is limited in river basins of Asia Pacific, a viable alternate is synthetic precipitation data, obtained from weather generators. The synthetic generation of precipitation data needs
precise rainfall statistics and a weather generator capable of simulating synthetic weather data.

The SWAT model includes a built-in weather generator (also called WXGN) that generates weather data wherever required (e.g., when data for some dates is missing in observed series), and SWAT modelers can use the readily available CFSR statistics dataset as input for WXGN. Even though CFSR statistics cover the entire globe, they are generated from combination of reanalysis data and satellite information, and CFSR's precipitation-related statistics have reportedly low accuracy levels for
the Asia Pacific region. The APHRODITE dataset can be a better alternative for estimating core precipitation statistics for the Asia Pacific region, as it was generated by interpolation of observed rainfall gages in the Asia Pacific region.

Given the prior successful applications of APHRODITE in the region, this study proposes the APWS weather statistics dataset for the Asia Pacific region, that combines precipitation weather statistics from APHRODITE with other weather statistics from CFSR. The APWS dataset is specifically designed to work with SWAT for generation of synthetic weather data, wherever
observed weather data is unavailable.

A comprehensive experimental (model-based) comparison of APWS and CFSR is also conducted in this study, that shows that APWS outperforms CFSR, to simulate hydrologic flows with SWAT in scenarios where observed precipitation data is missing. Both APWS and CFSR statistics are applied to SWAT models of two river basins in Asia, i.e., Narayani and Wangchhu. The APWS statistics clearly outperform CFSR in generating synthetic rainfall data (wherever observed data is missing) and
subsequently simulating the daily flows in both river basins, particularly for smaller independent sub-basins.

The APWS dataset is available via a web interface that has been developed for its public and easy access. Further investigations may be required to verify and improve the performance of APWS in other basins of region with contrasting climates. Hence, the authors encourage further testing of the APWS dataset in the Asia Pacific region, which has been prepared at a higher spatial resolution (0.25 *0.25°) than that of existing CFSR (0.38*0.38°) dataset.





## 7 Author contribution

UG and NS conceptualized the study. UG handled the APHRODITE data extraction and generation of weather statistics for entire Asia Pacific. NS developed the hydrological models for the study basins. TA developed the web platform and the automation of the hydrological models to check their performance using the weather statistics. UG drafted the manuscript and TA, NS and PD provided their comments and revision.

## 8 Acknowledgement

The required rainfall, temperature and flow data for Narayani river basin were acquired from RIMES project on "Development of End to End Flood Forecasting System and Decision Support System in Nepal", where the first author was working as a hydrologist previously. Special thanks go to Department of Hydrology and Meteorology, Nepal and Mr. Sangay Tenzin, Chief of Hydrology division, National Center for Hydrology and Meteorology, Bhutan.

## 9 Competing interests

The authors declare that they have no conflict of interest.

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



**Table 1 Weather statistics used by SWAT model to generate or fill missing weather inputs**

| No. | Variable name | Description | How does it affect weather generation in SWAT? |
|-----|---------------|-------------|------------------------------------------------|
| 1 | TITLE | Name of the weather station | Does not affect |
| 2 | WLATITUDE | Latitude of weather station in decimal degrees | Each subbasin in SWAT is assigned with closest weather generator based on latitude and longitude which changes all weather statistics |
| 3 | WLONGITUDE | Longitude of weather station in decimal degrees | |
| 4 | WELEV | Elevation of the weather station in meters | Rainfall and temperatures are defined accordingly for each elevation band depending upon the elevation of weather station and elevation of the weather statistics |
| 5 | RAIN_YRS | Number of years of weather data used to generate weather statistics | Maximum 0.5 hourly rainfall for the sub-basins are defined based upon the number of years |
| 6 | TMPMX | Average daily maximum temperature for a given month of all years | Needed to generate mean temperature at the center of basins when TLAPS parameter is considered for elevation bands. They are also used to estimate potential evapotranspiration and other weather variables |
| 7 | TMPMN | Average daily minimum temperature of a month for all years | |
| 8 | TMPSTDMX | Standard deviation of daily maximum temperature for a month in all years | The mean and standard deviation of temperatures are also used to find the amount of temperature to fill given the status of day (rainy/non-rainy) |
| 9 | TMPSTDMN | Standard deviation of daily minimum temperature for a month in all years | |





| 10 | PCPMM | Total precipitation for a month averaged for all years | The mean, standard deviation and skewness parameter are used to determine the amount of precipitation to fill for a rainy day |
|---|---|---|---|
| 11 | PCPSTD | Standard deviation of daily rainfall in a month for all years | |
| 12 | PCPSKW | Skew coefficient of daily rainfall in a month | |
| 13 | PR_W(1,n) | Probability of a wet day following dry day in "n" month for all years | Based upon the probabilities of a missing day, it will determine whether the day will be rainy or not for each month |
| 14 | PR_W(2,n) | Probability of wet day following wet day in "n" month for all years | |
| 15 | PCPD | Average number of rainfall days in a month | Used in PLAPS parameter to generate rainfall at each elevation band |
| 16 | RAINHHMX | Maximum half hour rainfall in a month for all years | Unknown |
| 17 | SOLARAV | Average solar radiation for a month for all years | Used in generation of series of solar radiation, dew point and wind speed to use for evapotranspiration calculation using Penman Monteith method |
| 18 | DEWPT | Average dew point temperature for a month for all years | |
| 19 | WNDAV | Average wind speed for a month for all years | |





Fig. 1 Generation of APWS dataset for Asia Pacific region



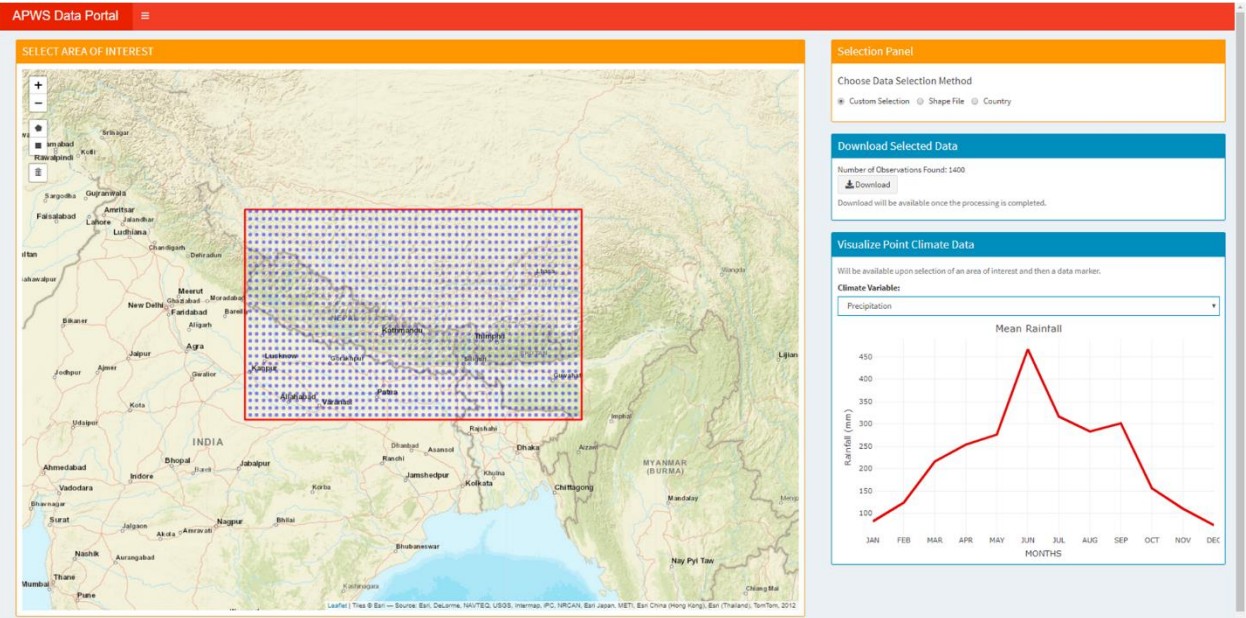


**Fig. 2 Web platform designed to disseminate APWS data in Asia Pacific basins**





Fig. 3 Performance evaluation of APWS for selected river basins of Asia







**Fig. 4 Location of the test basins in Asia Pacific region and their elevational information**




**Fig. 5 Comparison of distribution (top two rows; plotted over monthly rainfall time-series from years 1981-2007) and seasonality (bottom two rows) of mean monthly observed (Obs) rainfall (mm) with APHRODITE (APHRO) and CFSR rainfall series at selected stations of the study basins**


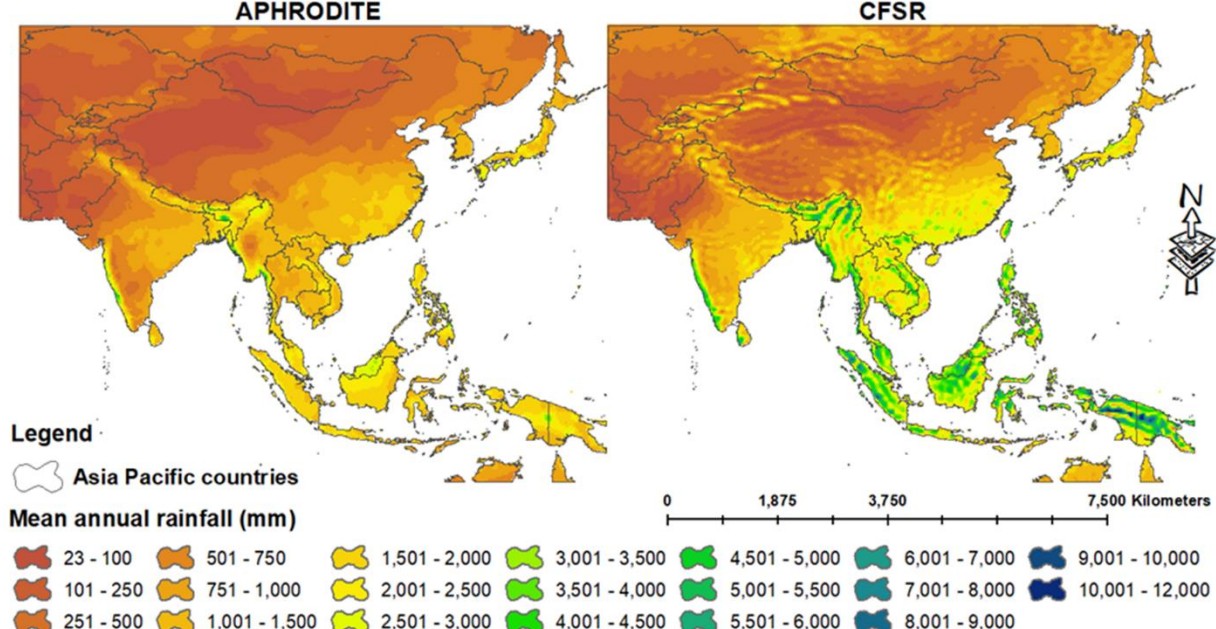

**Fig 6 Mean annual rainfall (mm) generated from APHRODITE and CFSR data for Asia Pacific region**



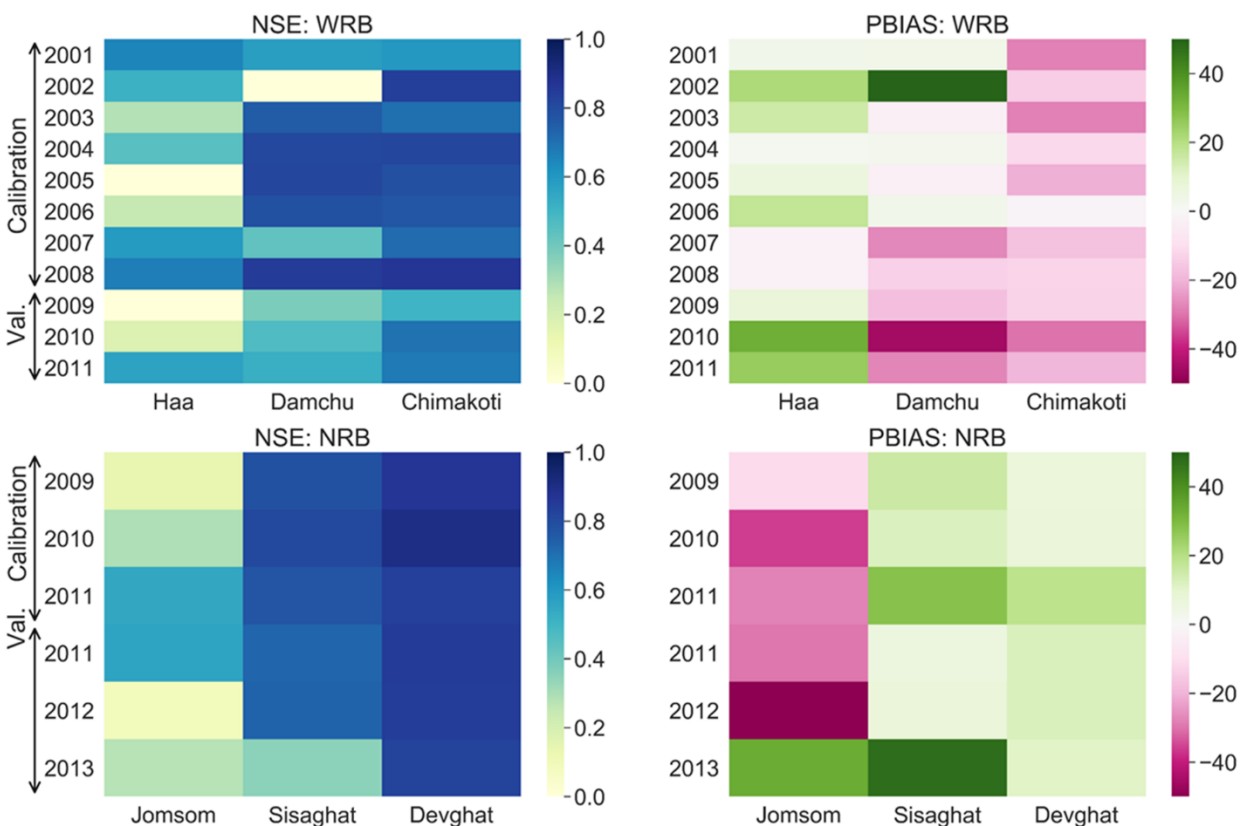

**Fig. 7 NSE (left column) and PBIAS (right column) computed from daily simulated and observed flows for each year in top, middle and lower parts (left to right of each plot) in WRB (top row) and NRB (bottom row)**

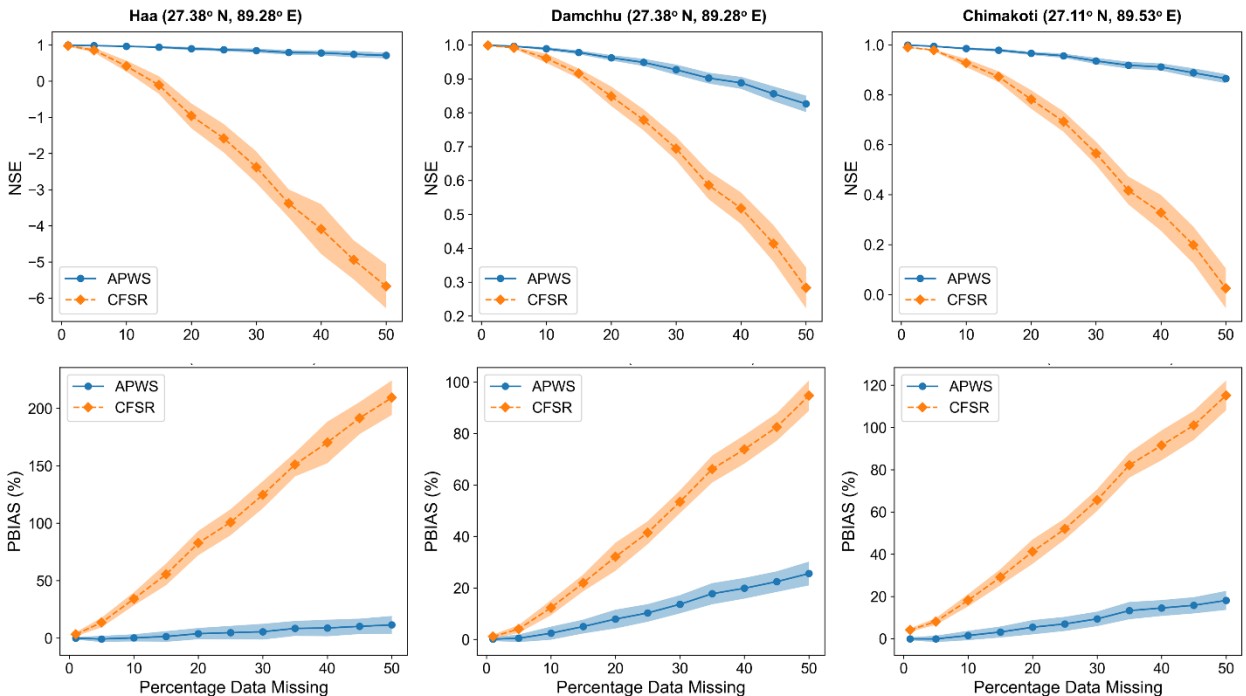

**Fig. 8 NSE and PBIAS in WRB computed using reference flows and simulated flows using weather statistics from APWS and CFSR for different scenarios of missing data (bands here represent the NSE and PBIAS values computed for 100 bootstraps) in WRB**



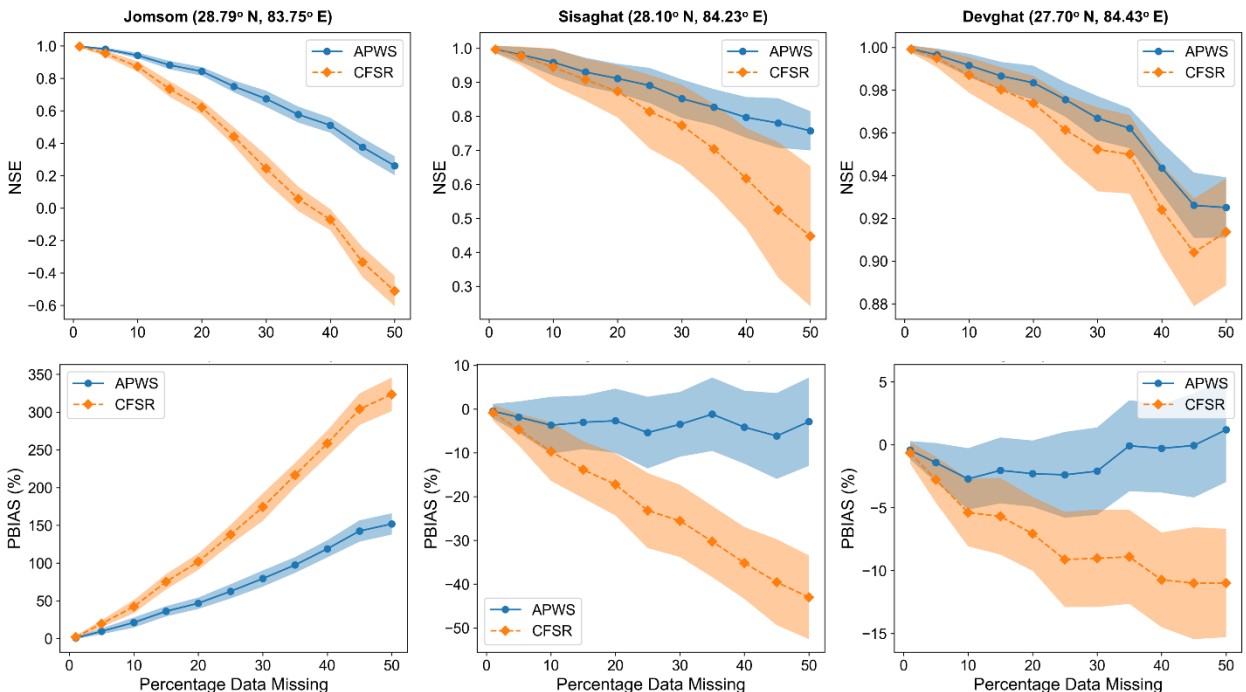

**Fig. 9 NSE and PBIAS in NRB computed using reference flows and simulated flows using weather statistics from APWS and CFSR for different scenarios of missing data (bands here represent the NSE and PBIAS values computed for 100 bootstraps) in NRB**

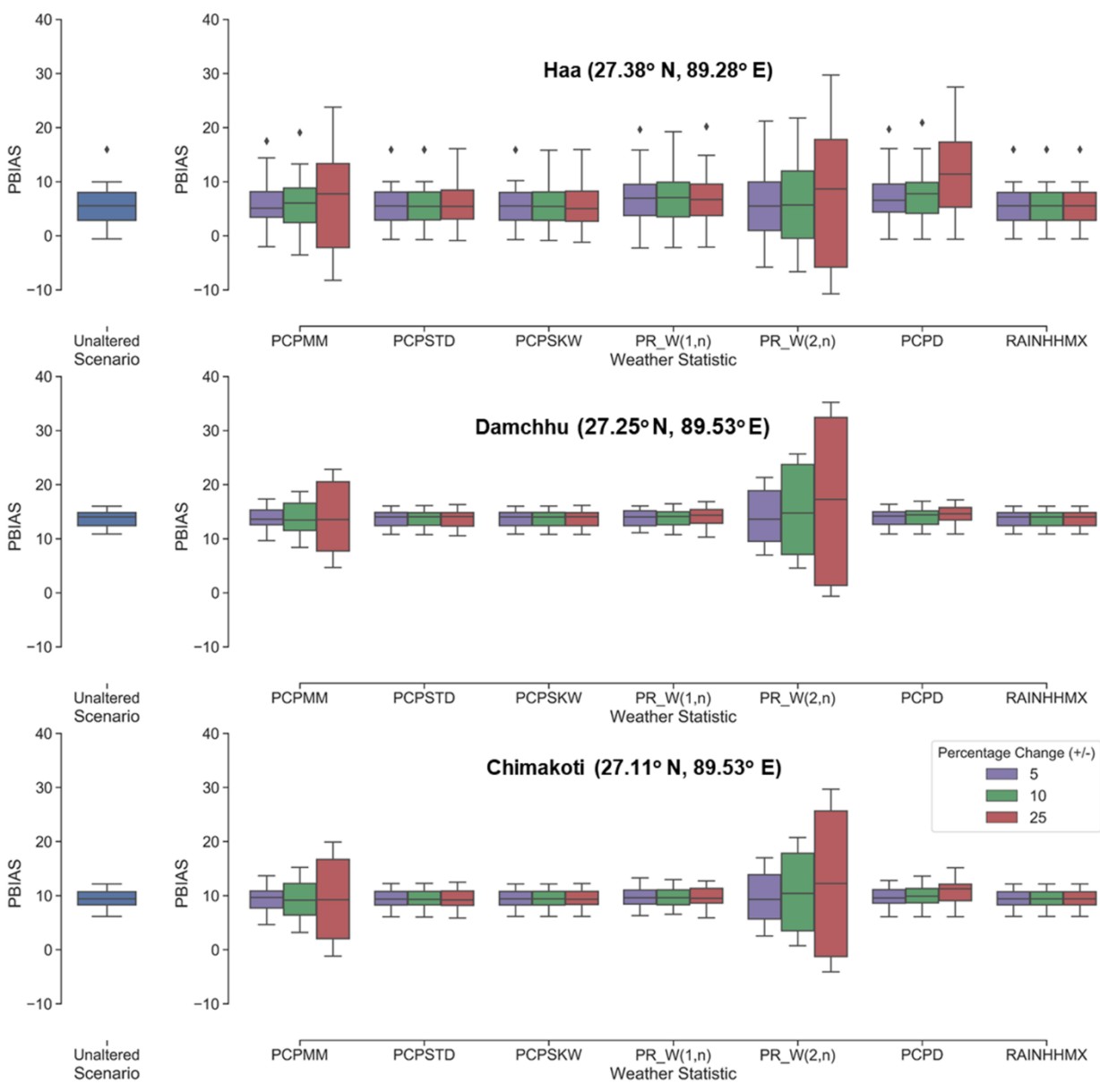

**Fig. 10 Sensitivity analysis of rainfall statistics in simulating daily flows at Haa, Damchu and Chimakoti flow stations in the Wangchhu river basin, Bhutan**