# Peer review of "Development of Asia Pacific Weather Statistics (APWS) dataset for use in Soil and Water Assessment Tool (SWAT) simulations"

_Earth System Science Data, 2019_

## Referee Comment (RC1) · Anonymous Referee #1 · 15 Dec 2019

The manuscript (essd-2019-178) aimed to developing a high-resolution Asia Pacific Weather Statistics (APWS) dataset in a format ready to be used in SWAT simulations. However, there have several issues in the ms as the requests of ESSD.

1. Authors should highlight why the APWS is a original research dataset? Because the current ms presented that the APWS dataset is only a repacking of APHRODITE for running the SWAT software. Besides, there has no description on the improvement of original dataset.

2. The evaluation of APWS dataset is not reasonable. The evaluation comparison between APWS and CFSR datasets is not necessary or a inferior way. Authors should

focus on the observed-based evaluations to indicate the original and quality-improved APHRODITE (i.e., APWS).

3. The construction of the ms should be improved. Current ms (introduction, method, result, and discussion) focused on the data repacking and devoted to perform the comparison with CFSR. Authors should reorganize the ms and focus on improving the quality of frequently-used datasets for running the SWAT.
* * *

---

## Referee Comment (RC2) · Anonymous Referee #2 · 15 Dec 2019

The manuscript is logically structured, and written very well with adequate contextualization, clarity in objectives, description of methodology and discussion of results. Authors have put huge efforts to address the needs of such ready-to-use database and to develop a web-based platform to disseminate the dataset. However, they have simply used available APHRODITE and CFSR reanalysis products to calculate the statistics, and named the product with data from two sources as the new dataset. Authors may consider clarifying whether it's good enough to consider this as new dataset. Furthermore, evaluation by comparing performance between APHRODITE and CFSR may not be convincing enough unless validated with observe datasets at several ground stations. Furthermore, it would be good to provide Author's view/recommendation on
the size of the basins that the data can be applied with reasonably good accuracy, and other considerations and/or limitations that the potential user should be aware of while using the dataset.

---

## Author Comment (AC1) · 14 Feb 2020

**Response to Anonymous Referee # 1:**

The authors would like to thank Referee # 1 on providing valuable feedback on the manuscript. We duly acknowledge the value of the comments of Referee # 1 and have incorporated changes in the manuscript and the supplementary section accordingly. The remaining portion of this document provides a specific account of the authors' responses to the comments provided by Referee # 1. Please note that comments of Referee # 1 are marked in red, and the corresponding responses are provided in black text.

The manuscript (essd-2019-178) aimed to developing a high-resolution Asia Pacific Weather Statistics (APWS) dataset in a format ready to be used in SWAT simulations. However, there have several issues in the ms as the requests of ESSD.

1. Authors should highlight why the APWS is an original research dataset? Because the current ms presented that the APWS dataset is only a repacking of APHRODITE for running the SWAT software. Besides, there has no description on the improvement of original dataset.

**Response**:  The authors would like to clarify that APWS is not a weather time-series product that is an improvement over APHRODITE, but rather a new weather statistics dataset that may be used with a stochastic weather generator, for generating synthetic weather data for hydrologic modeling. Moreover, the statistical data embedded in APWS is derived from existing weather products, i.e., APHRODITE and CFSR.

Rainfall is the primary driver of hydrological cycle in majority of river basins in Asia Pacific region. Precise and gap-free rainfall data is thus needed for robust hydrological model setup. Soil and Water Assessment Tool (SWAT) is a widely established hydrological model, which uses an inherent weather generator (WXGN) to fill gaps in meteorological inputs. It is also possible to generate entirely new series of synthetic weather such as rainfall, maximum and minimum temperatures, relative humidity, wind speed and solar radiation, based on user-defined weather statistics in the WXGN. Currently, river basins in the contiguous US benefit from meteorological gap filling via availability of a First Order US stations weather statistics database, which is in-built within SWAT model. For river basins outside USA, user needs to specify the statistics manually using long-term daily observed data (20 years or more), which is cumbersome and error-prone, and mostly unavailable in data-scarce regions. Alternatively, SWAT modelers may use the existing CFSR database available at 0.38-degree spatial resolution, for specification of weather statistics to be used with WXGN. However, CFSR statistics are computed using reanalysis CFSR daily data, which has been reported to have inferior performance compared to APHRODITE in several river basins of Asia Pacific (please see lines 137-141 of the revised manuscript and references cited therein). Thus, daily data series of the APHRODITE product from 1981-2007 (more than 20 years) are used to derive rainfall-related weather statistics for APWS, across the Asia-Pacific region at 0.25-degree resolution.

Since APHRODITE is a rainfall-only product, rainfall-related weather statistics for APWS are derived from it and remaining weather statistics (that are required by the WXGN weather generator; see Table 1 of main manuscript for description of the statistics dataset) are interpolated from the CFSR database. For the convenience of potential SWAT modelers, APWS data is prepared in SWAT-ready format and disseminated via a web-platform to ensure easy access for any region of interest. As such, no other weather statistics dataset employing observed rainfall is available at the presented resolution for Asia Pacific region and thus the authors believe the presented APWS is a novel and important dataset.

Evaluation of performance of APWS in the context of hydrologic modeling (in scenarios where some observed precipitation records are missing) now includes both a comparison against the CFSR statistics dataset and the statistics derived from original observed station weather data (please refer to Section 4 of

the revised manuscript (lines 190-207)). The superiority of APWS dataset over CFSR is evident from model performance at the selected two river basins of Asia, Narayani (Nepal) and Wangchhu (Bhutan) (see section 4.5, lines 279-341). Moreover, it is observed that performance of APWS is comparable to observed rainfall generated weather statistics.

Furthermore, the originality of APWS dataset has been highlighted in the revised manuscript at the end of Introduction section (e.g., see lines 98-99).

2. The evaluation of APWS dataset is not reasonable. The evaluation comparison between APWS and CFSR datasets is not necessary or in an inferior way. Authors should focus on the observed-based evaluations to indicate the original and quality-improved APHRODITE (i.e., APWS).

**Response**: The authors agree that hydrologic evaluation of APWS should include comparison against synthetic weather generation within SWAT, using rainfall statistics derived from observed data. Hence, in the revised manuscript, the evaluation has been extended to include missing data SWAT simulation scenarios that use synthetic weather generated from observed rainfall-based weather statistics. Consequently, for the two river basins selected for hydrologic modeling, i.e., Narayani (Nepal) and Wangchhu (Bhutan), 79 and 7 surface rainfall stations, respectively, were used to setup the hydrologic models. Observed rainfall data for these stations was used to derive rainfall statistics for this dataset (called OBS statistics dataset in the revised manuscript in Section 4), for use within the WXGN weather generator in SWAT.

Then, in order to hydrologically evaluate and compare performance of OBS, APWS and CFSR statistics, artificial gaps were created in observed rainfall data (with 1%-50% missing rainfall records) for both Narayani and Wangchhu SWAT models. It should be noted here that during simulation, SWAT uses its built-in weather generator (WXGN) to synthetically fill gaps in observed precipitation records, using a statistics dataset provided as inputs. In our hydrologic evaluation experiments (discussed in Section 4), we iteratively used OBS, APWS and CFSR statistics during SWAT simulations, for different scenarios where a percentage of actual precipitation records were missing. We subsequently compared the reconstructed rainfall simulated flows using OBS, APWS and CFSR statistics, with the SWAT simulation scenario with no missing rainfall data. Results revealed that APWS had similar performance to OBS and it outperformed CFSR statistics in reconstructing the rainfall and simulating the flows with them.

Since, many river basins in Asia Pacific are devoid of high-quality meteorological time-series data (i.e., longer, gap-free high-quality data at daily time-step), a high-quality weather statistics dataset like APWS is required for gap filling and synthetic weather generation using SWAT.

Based on the comments of the reviewer, we have revised the evaluation mechanism for APWS to better highlight the value of APWS. The revised evaluation is included in the revised manuscript in Section 4 (lines 190-207) and Section 4.5 (lines 278-341).

3. The construction of the ms should be improved. Current ms (introduction, method, result, and discussion) focused on the data repacking and devoted to performing the comparison with CFSR. Authors should reorganize the ms and focus on improving the quality of frequently used datasets for running the SWAT

**Response:** Following this suggestion from the reviewer, we have made significant changes to the structure and organization of the manuscript. Consequently, we have given more attention towards describing APWS and explaining how it is an improved weather statistics dataset for hydrologic modeling (using SWAT) for the Asia Pacific region.

The revised manuscript also highlights the fact that the comparison of APWS has not been done solely with CFSR statistics, but also with observed rainfall-based weather statistics. Structural revisions are made throughout the revised manuscript. However most key changes are made in Section 4 (lines 190-207) and Section 4.5 (lines 278-341).

---

## Author Comment (AC2) · 14 Feb 2020

**Response to Anonymous Referee # 2:**

The authors would like to thank Referee # 2 on providing valuable feedback on the manuscript. We duly acknowledge the value of the comments of Referee # 2 and have incorporated changes in the manuscript and the supplementary section accordingly. The remaining portion of this document provides a specific account of the authors' responses to the comments provided by Referee # 2. Please note that comments of Referee # 2 are marked in red, and the corresponding responses are provided in black text.

**Comment**: The manuscript is logically structured, and written very well with adequate contextualization, clarity in objectives, description of methodology and discussion of results. Authors have put huge efforts to address the needs of such ready-to-use database and to develop a web-based platform to disseminate the dataset.

**Response**: The authors thank the anonymous reviewer for his/her positive comments.

**Comment**: However, they have simply used available APHRODITE and CFSR reanalysis products to calculate the statistics and named the product with data from two sources as the new dataset. Authors may consider clarifying whether it's good enough to consider this as new dataset.

**Response**: The authors firmly believe that the Asia Pacific Weather Statistics (APWS) dataset presented in the manuscript, is indeed a new statistics dataset designed for SWAT models, that is derived from APHRODITE and CFSR products. We agree with the reviewer that APWS statistics are derived from existing weather products. However, the choice of statistics derived from each product (specifically derivation of rainfall statistics from APHRODITE), plays an important role in ensuring that the synthetic weather generated from APWS is more accurate than the weather generated from the existing statistics dataset available for the Asia Pacific region. We illustrate this in the context of hydrologic modeling using SWAT and our findings are discussed in Section 4.5 of the revised manuscript.

Rainfall is the primary driver of hydrological cycle in majority of river basins in Asia Pacific region. Precise and gap-free rainfall data is thus needed for robust hydrological model setup. Soil and Water Assessment Tool (SWAT) is a widely established hydrological model, which uses an inherent weather generator (WXGN) to fill gaps in meteorological inputs. It is also possible to generate entirely new series of synthetic weather such as rainfall, maximum and minimum temperatures, relative humidity, wind speed and solar radiation, based on user-defined weather statistics in the WXGN. Currently, river basins in the contiguous US are benefitted for meteorological gap filling by the availability of the First Order US stations weather statistics database, which is in-built within SWAT model. For river basins outside USA, user needs to specify the statistics manually using long-term daily observed data (20 years or more), which is cumbersome and error-prone, and mostly unavailable in data-scarce regions. Alternatively, the user can define such statistics using existing CFSR database available at 0.38-degree spatial resolution. However, CFSR statistics are computed using reanalysis CFSR daily data, which has been reported to have inferior performance compared to APHRODITE in several river basins of Asia Pacific (please refer 137-141 and references cited therein). Thus, daily data series of APHRODITE from 1981-2007 (more than 20 years) are used to derive rainfall-related weather statistics for APWS, across the Asia Pacific region at 0.25-degree resolution.

Since APHRODITE only includes daily rainfall series, rainfall-related weather statistics for APWS are derived from it and remaining weather statistics are interpolated from the CFSR product. For the convenience of potential SWAT modelers, APWS is disseminated in SWAT-ready format via a web-platform for any region of interest within Asia Pacific. As such, no other weather statistics dataset (to be

used for synthetic weather generation) employing observed rainfall is available at the presented resolution for Asia Pacific region.

Furthermore, the originality of APWS dataset has been highlighted in the revised manuscript at the end of Introduction section, along the lines 98-99.

**Comment**: Furthermore, evaluation by comparing performance between APHRODITE and CFSR may not be convincing enough unless validated with observe datasets at several ground stations.

**Response**: To further validate the APHRODITE and CFSR statistics at several ground stations, we chose 15 additional countries (36 meteorological stations) from the Asia-Pacific (see Figure S1) to evaluate the performance of observed (NOAA derived from https://www.ncdc.noaa.gov/cdo-web/datatools/findstation), APHRODITE and CFSR rainfall during 1981-2007. The list of stations selected, their basic information and their comparison statistics (summarized via Nash Sutcliffe Efficiency (NSE) and Percent-Bias (PBIAS)) with APHRODITE and CFSR are shown in the Table below (this Table is also included in revised Supplementary Document). Moreover, observed, APHORDITE-based and CFSR-based rainfall data are visually compared and presented in Figures A (cumulative mean monthly time-series) and B (cumulative monthly distribution) below (these Figures are also included in the revised Supplementary Document).

Table: List of rainfall stations used for comparison, along with their location information and statistics (NSE and PBIAS) against monthly NOAA observed rainfall during 1981-2007

| Stations | Country | Lat | Long | Elevation | NSE | | PBIAS | |
|---|---|---|---|---|---|---|---|---|
| | | | | | APHRODITE | CFSR | APHRODITE | CFSR |
| AGARTALA | India | 23.88 | 91.25 | 16.00 | 0.85 | -1.73 | -10.50 | 55.5 |
| AHMADABAD | India | 23.07 | 72.63 | 55.00 | 0.87 | 0.84 | -31.8 | -0.90 |
| ALYANGULA POLICE | Australia | -13.85 | 136.42 | 20.00 | 0.75 | 0.90 | -32.1 | 11.20 |
| AMBON PATTIMURA | Indonesia | -3.70 | 128.08 | 12.00 | 0.57 | 0.59 | -8.00 | -15.2 |
| AMRITSAR | India | 31.71 | 74.80 | 230.40 | 0.50 | 0.51 | -45.30 | -46.4 |
| ANQING | China | 30.53 | 117.05 | 20.00 | 0.96 | 0.90 | -6.2 | 0.70 |
| ANYANG | China | 36.05 | 114.40 | 64.00 | 0.96 | 0.91 | -9.10 | -11.8 |
| ARANYAPRATHET | Thailand | 13.70 | 102.58 | 49.00 | 0.88 | 0.76 | -19.9 | 11.30 |
| BAISE | China | 23.90 | 106.60 | 177.00 | 1.00 | 0.61 | 0.70 | 40.1 |
| BANG NA AGROMET | Thailand | 13.67 | 100.62 | 6.00 | -0.89 | -0.16 | -56.6 | -36.90 |
| BAR KHAN | Pakistan | 29.88 | 69.72 | 1098.00 | 0.45 | -0.52 | -16.80 | -49.7 |
| BATAM HANG NADIM | Indonesia | 1.12 | 104.12 | 24.00 | -2.11 | -1.35 | -64.6 | -50.30 |
| BAU BAU BETO AMBIRI | Indonesia | -5.47 | 122.62 | 2.00 | -28.00 | -276.93 | 83.40 | 309.9 |
| BINTULU | Malaysia | 3.20 | 113.03 | 5.00 | -1.28 | -6.16 | -27.20 | -47 |
| BRUNEI INTERNATIONAL | Brunei | 4.94 | 114.93 | 22.30 | 0.51 | -2.41 | -8.00 | -39.3 |
| CA MAU | Vietnam | 9.18 | 105.15 | 2.00 | 0.17 | -1.70 | 52.90 | 103.4 |
| DANANG INTERNATIONAL | Vietnam | 16.04 | 108.20 | 10.10 | 0.38 | -0.38 | 60.40 | 106.8 |
| DANIEL Z ROMUALDEZ | Philippines | 11.23 | 125.03 | 3.00 | 0.47 | -2.34 | 18.80 | 31 |
| DONGFANG | China | 19.10 | 108.62 | 8.00 | 0.95 | 0.69 | 11.30 | 38.6 |
| H AS HANANDJOEDDIN | Indonesia | -2.75 | 107.76 | 51.00 | 0.48 | 0.47 | -15.3 | 3.60 |
| HAMBANTOTA | Sri Lanka | 6.12 | 81.13 | 20.00 | 0.88 | 0.63 | -6.10 | -21.1 |
| K. PARAMATHY | India | 10.95 | 78.08 | 181.00 | 0.00 | 0.47 | -66.4 | -34.70 |
| KIUNGA W.O. | Papua New Guinea | -6.08 | 141.18 | 35.00 | -7.61 | -2.48 | -58.5 | -33.40 |
| KUANTAN | Malaysia | 3.62 | 103.22 | 16.00 | -0.11 | 0.57 | -44 | -9.50 |
| M.O. RANCHI | India | 23.32 | 85.32 | 652.00 | 0.90 | 0.74 | -24.7 | -4.80 |
| MADANG W.O. | Papua New Guinea | -5.22 | 145.80 | 4.00 | -2.76 | -1.28 | -53.3 | -35.40 |
| MALACCA | Malaysia | 2.26 | 102.25 | 10.70 | -0.40 | -1.70 | -21.5 | 15.30 |
| NABIRE | Indonesia | -3.37 | 135.50 | 6.10 | -78.54 | -136.19 | -70.40 | -93.2 |
| PADANG TABING | Indonesia | -0.88 | 100.35 | 3.00 | -0.27 | -1.93 | -20.90 | 38.1 |
| PENANG INTERNATIONAL | Malaysia | 5.30 | 100.28 | 3.40 | 0.01 | -7.83 | -31.10 | 96.4 |
| PHNOM PENH INTERNATIONAL | Cambodia | 11.55 | 104.84 | 12.20 | -9.85 | -42.30 | 127.80 | 324 |
| PHU QUOC | Vietnam | 10.22 | 103.97 | 4.00 | 0.60 | -0.49 | 31.30 | 63.2 |
| PORT MORESBY W.O. | Papua New Guinea | -9.38 | 147.22 | 48.00 | 0.84 | -0.07 | -20.40 | -56.1 |
| SAVANNAKHET | Laos | 16.55 | 104.77 | 155.10 | -6.87 | -137.00 | 66.40 | 416.2 |
| SINGAPORE CHANGI INTERNATIONAL | Singapore | 1.35 | 103.99 | 6.70 | 0.59 | -0.48 | -8.80 | 20.6 |
| SURIGAO | Philippines | 9.80 | 125.50 | 55.00 | 0.33 | 0.79 | 28.5 | 2.30 |

[Figure]

Figure A: Plots of cumulative monthly (averaged across years) OBS (black with circle), APHRODITE (green with triangle) and CFSR (red with cross) rainfall (mm) during 1981-2007

[Figure]

Figure B: Density distribution plots of cumulative monthly OBS (black), APHRODITE (green) and CFSR (red) rainfall during 1981-2007

The visual comparison of observed (OBS derived from the NOAA), APHRODITE and CFSR rainfall data, along with time series based statistics such as NSE and PBIAS clearly show that APHRODTE outperforms CFSR in multiple locations across the Asia-Pacific and thus validates the use of APHRODITE for deriving rainfall statistics for the APWS dataset. It is also worthwhile to note that countries like Indonesia, Papua New Guinea, Cambodia and Laos are exhibiting highest discrepancies (for both APHRODITE and CFSR) compared to the observed datasets. This is expected as these countries have scarce rain-gauge networks and thus are unable to adequately represent the climatology and a significant portion of observed data is missing. Rainfall products like APHRODITE make use of surface stations to interpolate a continuous grid of data. If the surface rainfall stations are scant or not included in the interpolation process of APHRODITE or CFSR, the generated data may also have huge discrepancies, compared with observed data. Thus, the user needs a careful scrutinization of data before its usage. The visual interface of APWS (https://hydrawater.shinyapps.io/APWS/) allows a user to visually analyze and validate the APWS data (via rainfall and temperature time-series plots) before potential use in synthetic weather generation and hydrologic modelling.

We have included the above detailed comparative analysis of observed rainfall data with APHRODITE and CFSR in the supplementary section of the revised manuscript and have also included a summary of our findings in lines 136-141 (Section 3.1) of the main manuscript.

**Comment**: Furthermore, it would be good to provide Author's view/recommendation on the size of the basins that the data can be applied with reasonably good accuracy, and other considerations and/or limitations that the potential user should be aware of while using the dataset.

**Response**: A section has been added in the revised manuscript wherein general recommendations regarding the usage of data have been given. Please refer to lines 329-341 of the revised manuscript for the following revision:

**Revision**: A preliminary analysis of six hydrological stations (3 in each basin) in this study suggests that smaller river basins (within few thousand square kilometers) are likely to benefit more from the developed APWS dataset (for e.g. refer to locations of Haa and Jomsom stations in Wangchhu and in Narayani river basins in Fig. 4 and check their performances in Fig. 7 and Fig. 8). First order river basins exhibit higher variability among the flows simulated by different weather statistics than the second and tertiary order river basins (again refer to Fig. 4 and Fig. 7 and Fig. 8). Hence, synthetic rainfall generation from a more accurate statistics dataset like APWS is recommended for first over basins.

Our analysis also indicates that, for the two study basins, observed precipitation data gaps in the range of 0-30% are can be adequately filled with synthetic data using APWS. Performance of SWAT deteriorates significantly if more than 30% of observed precipitation data is missing. Hence, it is recommended that APWS be used in SWAT scenarios, where up to 30% observed rainfall data is missing (for larger basins, even 50% missing data scenarios may be acceptable). It should also be noted that this percentage threshold (i.e., 30%) may be an over-estimation for highly arid basins, where typically, the entire annual rainfall occurs within a day or two.

---

## Author Comment (AC3) · 14 Feb 2020

**Development of Asia Pacific Weather Statistics (APWS) dataset for use in Soil and Water Assessment Tool (SWAT) simulations**

Uttam Ghimire, Taimoor Akhtar, Narayan Kumar Shrestha, Prasad Daggupati

College of Engineering and Physical Sciences, University of Guelph, Ontario, N1G2W1, Canada

5 *Correspondence to* Prasad Daggupati (pdaggupa@uoguelph.ca), +1-519-760-9299

**Abstract.**

The application of Soil and Water Assessment Tool (SWAT) for hydrological modelling in Asia Pacific region is immense. However, a robust modelling practice is often constrained by limited amount and quality of weather data. In such conditions, SWAT uses an inherent statistical weather generator to generate synthetic series of weather inputs for which, long-term precise

10 weather statistics are needed. This study presents a high-resolution Asia Pacific Weather Statistics (APWS) dataset in a format ready to be used in SWAT simulations.

The APWS dataset consists of rainfall statistics from Asian Highly Resolved Observational Data Integration Towards Evaluation of Water Resources (APHRODITE) project at 0.25° and remaining weather statistics from Climate Forecast System Reanalysis (CFSR) at 0.38°. The utility of APWS is evaluated by comparing it's performance with established CFSR statistics

15 and observed weather statistics (OBS) for daily flow simulation in two river basins of South Asia; Narayani in Nepal and Wangchhu in Bhutan. The comparison is done on different precipitation data availability scenarios, where for each scenario, a specified percentage of historical precipitation data is removed and replaced by synthetic precipitation data, generated by SWAT's inherent weather generator with weather statistics from i) OBS, ii) APWS and iii) CFSR independently.

Results indicate that performance of APWS is comparable to OBS and better than CFSR dataset in rainfall reconstruction for

20 hydrologic modelling, especially in the smaller sub-basins. Sensitivity analysis indicates that simulated hydrologic response of SWAT is highly sensitive to rainfall-based weather statistics like probability of wet day following wet day, mean monthly rainfall and number of rainy days. Hence, the use of highly accurate rainfall statistics is important for hydrologic modelling in data-scarce scenarios. These findings illustrate that APWS is a valuable dataset contribution for hydrological modelling 
[revised manuscript text omitted]

statistics dataset, in the context of synthetic weather generation and subsequent flow simulation in selected test basins. A high-
resolution weather statistics dataset at 0.25º is generated (hereafter named APWS dataset, i.e., Asia Pacific Weather Statistics
dataset) by combining rainfall statistics from APHRODITE and remaining weather statistics from nearest CFSR station at
0.38º spatial resolution and is made publicly accessible at https://hydra-water.shinyapps.io/APWS/ or
http://doi.org/10.5281/zenodo.3460766 (Ghimire et al., 2019) in SWAT ready format. Two river basins, Narayani in Nepal

and Wangchhu in Bhutan are selected as test basins to compare the performance of APWS against OBS and CFSR dataset, in weather generation and flow simulation for different missing percentages of rainfall. The presented APWS warrants originality,

100    since no other such weather statistics datasets are publicly available (that are designed for use within weather generators), where precipitation statistics are derived from observed rainfall at 0.25-degree resolution for entire Asia Pacific region in a SWAT-readable format.

**2 The SWAT weather generator statistics data structure**

[revised manuscript text omitted]

*[Fig. 2 about here]*

**4 Performance evaluation of APWS dataset**

In order to evaluate the performance of APWS dataset in effective hydrological simulation using SWAT in the Asian region, we used APWS for synthetic weather data generation for SWAT models of two river basins: Narayani (NRB) in Nepal and Wangchhu (WRB) in Bhutan. Figure 3 provides an overview of the design of our performance evaluation experiment. Section 4.1 presents the acquisition of data for selected river basins. Section 4.2 presents a brief comparison of monthly normals of observed, CFSR and APHRODITE rainfall. We then develop, calibrate and validate SWAT models of the Narayani and Wangchhu basins (see Sect. 4.3). The calibrated SWAT models use historical rainfall records at multiple stations during model development and calibration. In order to compare the performance (in the context of hydrologic modelling) of precipitation statistics of APWS, the default CFSR (normally used in SWAT) and statistics derived from observed rainfall (also called OBS), we develop alterations (also called 'missing precipitation data' scenarios) of the historical precipitation dataset where, in each scenario a specified percentage of historical data is missing (the missing days are randomly selected; discussed in Sect. 4.4). The SWAT models are then run with 'missing precipitation data' scenarios using i) observed rainfall-based weather statistics (OBS), ii) CFSR and iii) APWS weather statistics (to generate synthetic precipitation records for missing precipitation days; also called reconstructed rainfall) and hydrological simulations using the reconstructed rainfall records are compared. The flows simulated using reconstructed rainfall (from OBS, APWS and CFSR) are compared (see Sect. 4.5 for details) with flows simulated with unaltered rainfall (i.e., 0% missing data), as presented in methodological framework of Fig. 3. The OBS statistics are included in this study for a scenario, when modeler can generate the required weather statistics from long-term observed daily weather information and use them in SWAT. This also provides a benchmark for the performance evaluation of gridded products like CFSR and APWS.

*[Fig. 3 about here]*

**4.1 Data acquisition for selected basins**

The required rainfall, temperature and flow observations at daily timestep are acquired through Regional Integrated Multi Hazard Early Warning Systems (RIMES) center, Thailand and National Center of Meteorology and Hydrology (NCHM), Bhutan. Two river basins, Narayani (hereafter named NRB) in Nepal (36,000 sq.km) and Wangchhu (hereafter named WRB) in Bhutan (3,600 sq. km) are considered in this study to compare performance of APWS, CFSR and OBS statistics in weather generation. The location of NRB and WRB in south Asia, along with their topographical information and the flow stations considered in this study is presented in Fig. 4.

*[Fig. 4 about here]*

The NRB consists of 79 rainfall, 36 temperature and 3 flow stations as presented in Fig. S5. Similarly, the WRB has 7 rainfall, 7 temperature and 3 flow stations, as presented in Fig. S6. The meteorological and flow data are available for the years 2008-2014 in the NRB and 2000-2014 in the WRB respectively.

**4.2 Comparison of rainfall normals**

[revised manuscript text omitted]

missing precipitation scenarios discussed in Sect. 3.4) with ii) unaltered rainfall simulated flows (also called baseline flows as discussed in Sect. 4.4. The results of performance evaluation for the WRB is presented in Fig. 7, where the lines represent average NSE and PBIAS values and shaded areas represent their standard deviations, for each of OBS, APWS and CFSR simulated daily flows.

*[Fig. 7 about here]*

Results for WRB stations clearly show that accuracy of weather statistics is of paramount importance in filling the rainfall series and subsequently, in accurate simulation of hydrologic flows. The OBS and APWS datasets are found to have similar performance. Moreover, APWS clearly outperforms CFSR since both NSE and PBIAS values for APWS remain reasonable even with 50% missing precipitation data. Moreover, the difference in performance of the OBS, APWS and CFSR statistics is more significant in smaller sub-basins located in upper parts of the basin (e.g., represented by the Haa station in left-most panels of Fig. 7), compared to the lower parts. A reason for this difference could be the subsequent dampening of the missing rainfall events, as the flow progresses downstream. The smaller sub-basins located in the upper parts of the study basins are more flashy in nature compared to the lower sub-basins, which has been established to negatively impact the hydrological model performance (Poncelet et al., 2017). The performance of hydrological models is also generally better at the downstream locations and increases with size of basins (Merz et al., 2011;Van Esse et al., 2013)

A significant deviation of NSE is observed for all flow stations in WRB, when CFSR weather statistics are used to fill the missing rainfall series in the WRB. The rainfall statistics of CFSR were significantly different than observed and APHRODITE data for the basin, as evident from Fig. 5. This is likely to yield large errors from the baseline simulated flows (i.e., without missing precipitation data) when CFSR is used to fill the missing rainfall series. The biased nature of CFSR in WRB is also evident from the PBIAS computed at its flow stations. The biases aggregate more than 50% in all stations, when 20% or less rainfall data is missing, and the weather generator with CFSR statistics is used to generate synthetic rainfall data for missing days. The APWS however is found to have almost similar performance to that of OBS, as the differences between the observed and APHRODITE rainfall were also minimal as evident in Fig. 5. A similar performance of APWS with OBS and overall relative superiority (of both datasets) over CFSR is also evident for NRB in Fig. 8, wherein NSE and PBIAS indices for APWS and OBS datasets are clearly better than the corresponding NSE and PBIAS indices for CFSR.

*[Fig. 8 about here]*

Results of NRB, as presented in Fig. 8, also depict that the size and location of a sub-basin has a significant impact on performance of weather statistics in simulating hydrologic flows, i.e., smaller sub-basins that are located in upper parts of basins, and have no contributions from other tributaries, tend to be heavily reliant on accurate observed weather data for accurate hydrologic simulation. The Jomsom hydrological station located in the northernmost part of the NRB (Fig. 8, left-most panels) is part of a small sub-basin of NRB and is devoid of contribution from other tributaries in the basin. Moreover, the sub-basin that Jomsom drains has an arid climatology, with a mean annual rainfall of around 350 mm (as presented in Fig. S7). Arid basins have been known to have lower model efficiency compared to wet basins (Poncelet et al., 2017). Hence, the NSE and PBIAS values at Jomsom, become significantly worse (compared to the other stations located in

lower parts of the basin), as the percentage missing data value increases slightly. In arid and semi-arid sub-basins, total rainfall is mostly contributed by rainfall events rather than rainfall seasonality, due to which even a smaller percentage of missing data concentrated around such events is likely to deteriorate the hydrological model performance. The use of weather generator to reconstruct the missing rainfall is thus likely to change the rainfall sequence in such basins thus degrading the performance of weather generators significantly even for few missing events. Similarly, as the size of basin increases and as we approach the lower parts of NRB where rainfall volume is significant, the reduction in performance of the weather statistics is gradual. Both APWS and CFSR datasets perform adequately in NRB for stations located in the lower part of the river basin (see results for Sisaghat and Devghat in Fig. 8). However, performance of APWS is slightly better for these stations as well and almost matches to that of OBS. Overall, a consensus could be derived from both study basins that performance of APWS is similar to that of OBS and better than CFSR statistics, in terms of deriving synthetic rainfall data for missing days at observed weather stations and, subsequently, in simulating hydrologic flows under limited precipitation data availability scenarios.

A preliminary analysis of six hydrological stations (3 in each basin) in this study suggests that smaller river basins (within few thousand square kilometers) are likely to benefit more from the developed APWS dataset (for e.g. refer to locations of Haa and Jomsom stations in Wangchhu and in Narayani river basins in Fig. 4 and check their performances in Fig. 7 and Fig. 8). First order river basins exhibit higher variability among the flows simulated by different weather statistics than the second and tertiary order river basins (again refer to Fig. 4 and Fig. 7 and Fig. 8). Hence, synthetic rainfall generation from a more accurate statistics dataset like APWS is recommended for first over basins.

Our analysis also indicates that, for the two study basins, observed precipitation data gaps in the range of 0-30% are can be adequately filled with synthetic data using APWS. Performance of SWAT deteriorates significantly if more than 30% of observed precipitation data is missing. Hence, it is recommended that APWS be used in SWAT scenarios, where up to 30% observed rainfall data is missing (for larger basins, even 50% missing data scenarios may be acceptable). It should also be noted that this percentage threshold (i.e., 30%) may be an over-estimation for highly arid basins, where typically, the entire annual rainfall occurs within a day or two.

[revised manuscript text omitted]

---

## Author Comment (AC4) · 14 Feb 2020

The comment was uploaded in the form of a supplement:
https://www.earth-syst-sci-data-discuss.net/essd-2019-178/essd-2019-178-AC4-
supplement.pdf